# Hough Transform and Clustering for a 3-D Building Reconstruction with Tomographic SAR Point Clouds

**DOI:** 10.3390/s19245378

**Published:** 2019-12-05

**Authors:** Hui Liu, Lei Pang, Fang Li, Ziye Guo

**Affiliations:** School of Electrical and Information Engineering, Beijing University of Civil Engineering and Architecture, Beijing 100044, China; lifang@bucea.edu.cn (F.L.); panglei@bucea.edu.cn (L.P.); guoziye97@163.com (Z.G.)

**Keywords:** TomoSAR cloud point, supervised algorithm, unsupervised clustering, 3-D reconstruction, Hough transform, machine learning

## Abstract

Tomographic synthetic aperture radar (TomoSAR) produces 3-D point clouds with unavoidable noise or false targets that seriously deteriorate the quality of 3-D images and the building reconstruction over urban areas. In this paper, a Hough transform was adopted to detect the outline of a building; however, on one hand, the obtained outline of a building with Hough transform is broken, and on the other hand, some of these broken lines belong to the same segment of a building outline, but the parameters of these lines are slightly different. These problems will lead to that segment of a building outline being represented by multiple different parameters in the Hough transform. Therefore, an unsupervised clustering method was employed for clustering these line parameters. The lines gathered in the same cluster were considered to correspond to a same segment of a building outline. In this way, different line parameters corresponding to a segment of a building outline were integrated into one and then the continuous outline of the building in cloud points was obtained. Steps of the proposed data processing method were as follows. First, the Hough transform was made use of to detect the lines on the tomography plane in TomoSAR point clouds. These detected lines lay on the outline of the building, but they were broken due to the density variation of point clouds. Second, the lines detected using the Hough transform were grouped as a date set for training the building outline. Unsupervised clustering was utilized to classify the lines in several clusters. The cluster number was automatically determined via the unsupervised clustering algorithm, which meant the number of straight segments of the building edge was obtained. The lines in each cluster were considered to belong to the same straight segment of the building outline. Then, within each cluster, which represents a part or a segment of the building edge, a repaired straight line was constructed. Third, between each two clusters or each two segments of the building outline, the joint point was estimated by extending the two segments. Therefore, the building outline was obtained as completely as possible. Finally, taking the estimated building outline as the clustering center, supervised learning algorithm was used to classify the building cloud point and the noise (or false targets), then the building cloud point was refined. Then, our refined and unrefined data were fed into the neural network for building the 3-D construction. The comparison results show the correctness and the effectiveness of our improved method.

## 1. Introduction

Reconstruction of three-dimensional (3-D) building models in urban areas has been a hot topic in remote sensing, photogrammetry, and computer vision for more than two decades [1]. Tomographic synthetic aperture radar (TomoSAR) is a remote sensing technique that extends the conventional two-dimensional (2-D) SAR imaging principle to three-dimensional (3-D) imaging [2] (see Figure 1). SAR has the ability to work all day due to the active emission of signals. Moreover, SAR is almost independent of weather conditions because of the use of microwaves in radar signals, which is a major advantage when compared to sensors in the visible or infrared spectrum [3,4]. Very high resolution (VHR) satellite SAR imagery nowadays offers sub-meter resolution and TomoSAR techniques introduce the idea of synthetic aperture to the elevation, which make it possible to reconstruct 3-D building models from SAR images. It produces 3-D point clouds with unavoidable noise that seriously deteriorates the quality of 3-D imaging and the reconstruction of buildings over urban areas [5]. SAR is capable of assessing the deformation of the ground and buildings in the order of centimeters and millimeters due to its coherent nature and short wavelengths, and it supports various important application scenarios, such as damage assessment in urban areas after natural disasters [6]. Spaceborne TomoSAR are particularly suited for the long-term monitoring of such dynamic processes [7]. TomoSAR technology has a very important role in terrain monitoring and building deformation detection, especially for huge manmade facility-deformation monitoring in the future. In an urban remote sensing, building information retrieval and reconstruction from SAR images have been extensively investigated [8]. In recent years, scholars mainly focused on TomoSAR 3-D imaging algorithms, which can be roughly divided into three categories: backward projection [9], spectral estimation [10], and compressive sensing [11,12,13,14,15,16,17,18,19,20,21]. TomoSAR 3-D imaging algorithms, such as compressive sensing and spectral estimation-based multiple signal classification (MUSIC), are relatively mature at present and it produces 3-D point clouds with unavoidable noise that seriously deteriorates the quality of 3-D imaging and the reconstruction of buildings over urban areas. The point cloud obtained by these methods cannot be directly applied. Therefore, the question regarding how to reconstruct the building from the point cloud submerged by noise is an urgent problem in the TomoSAR research field.

Depending on, or in combination with, some other means [21,22], the reconstruction of buildings can be carried out. The detection and reconstruction of the building radar footprints from single very high resolution SAR images [6] are automatically detected and reconstructed using the 2-D and 3-D building shape from Spaceborne TomoSAR point clouds using a region-grow algorithm [22]. Today, machine learning and deep learning are widely spread [23] in the remote sensing process. Recurrent 3-D fully convolutional networks [23] are used in hyperspectral images for changing detection. A deep recurrent neural network [24] is utilized for the agricultural classification of SAR images. A neural network [5] is introduced in the 3-D reconstruction from a TomoSAR point cloud, but if the data is not cleaned well, the result will not be satisfactory. 

Toward solving this issue, a combined method with Hough transform and clustering was proposed in this study for filtering these interferences and training the continuous outline of a building in cloud points. The Hough transform was adopted to detect the outline of a building; however, on one hand, the obtained outline of a building from a Hough transform is broken, and on the other hand, some of these broken lines belong to the same segment of a building outline, but the parameters of these lines are slightly different. These problems will lead to an issue where a segment of a building outline is represented by multiple different parameters in the Hough transform. Therefore, an unsupervised clustering method was employed for clustering these line parameters. The lines gathered in the same cluster were considered to correspond to the same segment of a building outline. In this way, different line parameters corresponding to a segment of a building outline were integrated into one and then the continuous outline of a building in cloud points was obtained. Steps of the proposed data processing method were as follows. First, the Hough transform is made use of for detecting the lines on the tomography plane in TomoSAR point clouds. These detected lines lay on the outline of the building, but they were broken due to the density variation of point clouds. Second, the detected lines using the Hough transform were grouped as a date set for training the building the outline. Unsupervised clustering was utilized to classify the lines in several clusters. The cluster number was automatically determined using the unsupervised clustering algorithm, which means the number of straight segments of the building edge was obtained. The lines in each cluster were considered to belong to the same straight segment of the building outline. Then, within each cluster, which represented a part or a segment of the building edge, a repaired straight line was constructed. Third, between each two clusters or each two segments of the building outline, the joint point was estimated by extending the two segments. Therefore, the building outline was obtained as completely as possible. Finally, taking the estimated building outline as the clustering center, a supervised learning algorithm was made use to classify the building cloud point and the noise (or false targets), and then the building cloud point was refined. Then, our refined and unrefined data were feed into the neural network for building the 3-D construction. The comparison results showed the correctness and the effectiveness of our improved method.

## 2. Framework of the TomoSAR 3-D Building Reconstruction

The TomoSAR 3-D building reconstruction framework is illustrated in Figure 2. In the process of data processing, the building outline varied between different coordinate systems and Figure 3 shows the building outline modification chain in the TomoSAR 3-D reconstruction processing. The purpose of tomographic SAR is to inverse the reflectivity profile for each azimuth-range pixel (see Figure 1) and then get the 3-D building point clouds. 

Compressive sensing of a sparse signal is a common method to obtain tomographic SAR 3-D point clouds, but the compressive sensing imaging of tomographic SAR is used just to complete a partial inversion process of the 3-D building reconstruction as shown in Figure 4. TomoSAR 3-D imaging algorithms, such as compressive sensing and spectral estimation-based multiple signal classification (MUSIC), are relatively mature at present, but these methods produce 3-D point clouds with unavoidable noise that seriously deteriorates the quality of 3-D imaging and the reconstruction of buildings over urban areas. The point cloud obtained using these methods cannot be directly applied. Therefore, the question regarding how to reconstruct the building from the point cloud submerged in noise is an urgent problem in the TomoSAR research field.

## 3. 3-D Building Reconstruction from TomoSAR Point Clouds

For simplicity, in the following analysis, we assumed that the input point clouds of our method were projected from the range direction to the ground-range using the method described in Zhou et al. [5]. The schematic geometry of one single target building is illustrated in Figure 5. Considering the side-look imaging principle of TomoSAR, we made a simplifying assumption that the SAR did not penetrate the building and ground, such that the surface scatterers could be “seen” only from one side of the building [5]. Then, we directly used these surface scatterers to represent the building structure. An example of a three-dimensional reconstruction of point clouds would return all blue dots in Figure 5 to the red line as much as possible, and remove other false targets.

### 3.1. Hough Transform in the Tomographic Plane of 3-D Point Clouds

Typical compressive sensing (CS) algorithms, such as orthogonal match pursuit (OMP) and regularized OMP (ROMP), are employed to inverse the scatterers on the building surface. However, due to the existence of noise, there can still be a few false targets and outliers scattered in a disorderly manner in the tomographic plane, which caused the building to be submerged in these disorganized false targets. Therefore, it is necessary to detect the outline of the building in the tomography plane. Generally, the outline of a regular building in the tomography plane is usually composed of several straight connected line segments. The commonly used method of line detection is a Hough transform, which is widely used in computer vision and pattern recognition. Here is the principle of straight-line detection with a Hough transform. This algorithm is essentially a voting process where each point belonging to the patterns votes for all the possible patterns passing through that point. These votes are accumulated in an accumulator array called bins, and the pattern receiving the maximum votes is recognized as the desired pattern [25].

Given an N×N binary edge image, straight lines are defined as:(1)ρ=xcosθ+ysinθ,
where (x,y) is the measurement of the position in x-y coordinates, θ(−π/2≤θ<π/2) denotes the angle that the norm line makes with the x-axis and ρ($-N\leq \rho \leq \sqrt{2}N$) is the norm distance from the origin to the line. As shown in Figure 6, (x,y) denotes the coordinates of the red points, ρ and θ are defined as the parameter space of the Hough transform. The straight line is mapped to a point in the parameter space of the Hough transform. 

For example, every straight line through (x,y), such as the red dot (see Figure 6) in the x-y coordinates is mapped to a point in the parameter space of the Hough transform. Then, all the possible straight lines through the red dot in x-y coordinates will form a curved line, such as the red dashed curve in the parameter space of the Hough transform. For all parameter cells (ρ,θ), the Hough transform algorithm calculates the parameter values and accumulates the pixels that drop in the parameter cells (ρ,θ). If there are enough pixels, which are mapped to a parameter cells (ρ,θ), then (ρ,θ) is determined as a line in the x-y coordinates, and if not, it is determined as noise. 

The Hough transform computation consists of three parts: (1) calculating the parameter values and accumulating the pixels in the parameter space of the Hough transform, (2) finding the local maximums that represent line segments, and (3) extracting the line segments using the knowledge from the maximum positions. It visits each pixel of the image once.

### 3.2. Unsupervised and Supervised Clustering

After the Hough transform, the building outline is initially estimated in the tomography plane. However, on one hand, the detected lines were often broken due to the existing noise or the different density of the point cloud, as shown in Figure 7 (red line segments). On the other hand, some of these broken lines belonged to the same segment of a building outline, but the parameters of these lines were slightly different. Taking the Hough transform parameter (ρ,θ) as the data features, which are shown in Table 1, and utilizing a K-means clustering method, which is a typical unsupervised learning algorithm, the detected lines (red line segments) were grouped in several clusters, as illustrated in Figure 7b (each color represents a cluster). Here, we used the parameter distance for clustering. The parameter distance can be written as:(2)$$R_{k,l}=\sum_{k,l}{\left(\rho_{k}-\rho_{l}\right)}^{2}+{\left(\theta_{k}-\theta_{l}\right)}^{2},\;(l,k=1,2,\cdots \mathrm{and}\;l\neq k).$$

For example, in Table 1, l,k=1,2,⋯,12 and l≠k. If $R_{k,l}\leq \varepsilon$ (ε is very small), these two lines belong to a same straight segment. However, one should be careful when treating situations such as $\left|\theta_{k,l}\right|\approx 90^{\circ }$, $\left|\theta_{k,l}\right|\approx 0^{\circ }$, and $\left|\rho_{k,l}\right|\approx 0$. This is caused by the characteristics of the Hough transform; here, we modified the distance formula to become:(3)$$R_{k,l}=\min\left(\left\{ \begin{array}{ll} \sum_{k,l}{\left(\rho_{k}+\rho_{l}\right)}^{2}+{\left(\theta_{k}+\theta_{l}\right)}^{2}, & if\;\left|\theta_{k,l}\right|\approx 90^{\circ }\\ \sum_{k,l}{\left(\rho_{k}-\rho_{l}\right)}^{2}+{\left(\theta_{k}+\theta_{l}\right)}^{2}, & if\;\left|\theta_{k,l}\right|\approx 0^{\circ }\\ \sum_{k,l}{\left(\frac{\left|\rho_{k}\right|+\left|\rho_{l}\right|}{2}\right)}^{2}+{\left(\left|\theta_{k}+\theta_{l}\right|-90\right)}^{2}, & if\;\left|\rho_{k,l}\right|\approx 0\\ \sum_{k,l}{\left(\rho_{k}-\rho_{l}\right)}^{2}+{\left(\theta_{k}-\theta_{l}\right)}^{2}, & others \end{array}\right.\right),\;(l,k=1,2,\cdots \mathrm{and}\;l\neq k)$$

For example, in Table 1, lines 7 and 9 belong to the same straight segment l,k=1,2,⋯,12 and l≠k. For the situation $\theta_{k,l}\approx \pm 90^{\circ }$, which corresponds to horizontal lines in the image, if there is a small error in the Hough transform, it means that $\theta_{k}\approx -\theta_{l}$ and $\rho_{k}=-\rho_{l}$. For the situation $\theta_{k,l}\approx \pm 0^{\circ }$, which corresponds to vertical lines in the image, if there is a small error in the Hough transform, it means $\theta_{k}\approx -\theta_{l}\approx \pm 0^{\circ }$ and $\rho_{k}=\rho_{l}$. For the situation $\left|\rho_{k,l}\right|\approx 0$, which corresponds to slope lines passing through the origin of the coordinates in the image, if there is a small error in the Hough transform, it means $\left|\theta_{k}+\theta_{l}\right|\approx 90^{\circ }$.

Each cluster belongs to the same segment of the building outline. As shown in Figure 7b, yellow lines, green lines, and red lines represent the lines in different segments of the building outline. Within each cluster or each segment, the parameter (${\hat{\rho }}_{i},{\hat{\theta }}_{i}$, i=1,2,⋯,n), which represents a segment of the building, outline is estimated. The parameter n is the number of clusters and it is obtained from the procedure of unsupervised clustering. Using the estimated parameter (${\hat{\rho }}_{i},{\hat{\theta }}_{i}$, i=1,2,⋯,n), the continuous building outline function $\hat{f}(x)$ is established. It can be written as:(4)$$\hat{f}(x)=\sum_{i=1}^{n}{\hat{a}}_{i}x+{\hat{b}}_{i},\;i=1,2,\cdots ,n,$$
where the estimated parameters ${\hat{a}}_{i}(i=1,2,\cdots ,n)$ and ${\hat{b}}_{i}(i=1,2,\cdots ,n)$ are:(5)$$\left\{ \begin{array}{l} {\hat{a}}_{i}=\frac{{\hat{\rho }}_{i}}{\sin{\hat{\theta }}_{i}}\\ {\hat{b}}_{i}=\frac{\cos{\hat{\theta }}_{i}}{\sin{\hat{\theta }}_{i}} \end{array}\right.,\;i=1,2,\cdots ,n.$$

Then, taking the estimated building outline function $\hat{f}(x)$ as the clustering center, supervised clustering was used to separate the false targets and the noise from the building point cloud. The method can be expressed as:(6)$$g(x,y)=\left|\hat{f}(x)-y\right|,$$
where (x,y) is the position of the arbitrary pixels in binary image. If (x,y) is very close to the building outline, Equation (6) almost equals zero, and the pixels are determined as the building point cloud. Conversely, the point is determined as noise or a false target. Figure 7c,d exhibits the procedure of clustering.

## 4. Simulations and Experiments

Zhu et al. [26], Peng et al. [27], and Liang et al. [28] have focused on tomography SAR to make use of SAR data collected with the TerraSAR-X radar satellite, which is the first satellite in Germany. The TerraSAR-X radar satellite observes the earth at an orbital height of 514 km during the day and night. The resolution of the SAR data received by it is less than 1 m after 2-D SAR imaging. Budillon et al. [15] and Zhao et al. [29] studied the tomographic SAR imaging using Cosmo-SkyMed data. Due to insufficient data in our laboratory, we plan to use simulation data to verify the above theoretical derivation and the data processing flow.

### 4.1. Simulation Parameters 

Referring to the TerraSAR-X radar satellite, the simulation parameters of the SAR system are shown in Table 2. The simulation baselines and inclinations are shown in Table 3. According to the baseline parameters generated by the simulation, the maximum baseline B formed using 24 images was 970.593 m in the elevation direction. Using the baseline and radar parameters, we simulated datasets of 24 SAR images only along the range direction, which was about a building whose height was 150 m, as shown in Figure 8.

### 4.2. CS-Based TomoSAR Imaging

The most commonly used CS algorithms are utilized for TomoSAR imaging. Figure 9a,b exhibit the TomoSAR imaging results of the orthogonal matching pursuit (OMP) and regularized orthogonal match pursuit (ROMP). From these two figures, we can see that the point clouds of the building were submerged in noise and false targets.

### 4.3. Hough Transform Line Detection and Point Cloud Clustering

Tomographic imaging point cloud with ROMP was used to verity our method for the building reconstruction. First, the point cloud was transformed into a binary image, which can be seen in Figure 10a. Then, the Hough transform was operated on it and the result is shown in Figure 10b. Table 1 gives the parameters (ρ,θ) collected with the Hough transform, which can be present the detected broken line features. The detected lines are shown in Figure 10c.

Taking the data shown in Table 1 as the data set, we can see the data set with the lines had four features; using features *ρ* and *θ* of the Hough transform parameters, these samples of lines could be mainly grouped into three categories using unsupervised clustering. The clustering results are shown in Figure 11. In Figure 11a, the block of red represents the red lines; in Figure 11b, the yellow block represents the yellow lines and the green block denotes the green lines. In Figure 11b, we can see some broken lines lay on the building profile.

Within a cluster, using line features, such as start pixels and end pixels, we could get the repaired lines that were very close to the whole outline of the building, as shown in Figure 11c. The method is illustrated using Equation (7):(7)$$\underset{\hat{i},\hat{j}}{\max}\left|p_{i}-p_{j}\right|,(i,j=1,2,\cdots )s.t.\left\{ \begin{array}{l} {\hat{p}}_{nstart}=p_{\hat{i}}\\ {\hat{p}}_{nend}=p_{\hat{j}} \end{array}\right.,$$
where ${\hat{p}}_{nstart}$ and ${\hat{p}}_{nend}$ denote the start and end pixels of the repaired lines in each cluster, respectively. The aim was to collect these start and end pixels together within each cluster and redefine them as $p_{i}(i=1,2,\cdots )$, and then find the start pixel ${\hat{p}}_{nstart}$ and end pixel ${\hat{p}}_{nend}$ to repair the broken line segment using Equation (7). The repaired line segments are shown in Figure 11c (red line segments).

Then, between each two repaired line segments of the building outline, the estimated joint point ${\hat{p}}_{c}({\hat{x}}_{c},{\hat{y}}_{c})$ in Equation (8) was estimated by extending each two repaired line segments:(8)$$\left\{ \begin{array}{l} {\hat{x}}_{c}=-\frac{{\hat{b}}_{i}-{\hat{b}}_{j}}{{\hat{a}}_{i}-{\hat{a}}_{j}}\\ {\hat{y}}_{c}=-{\hat{a}}_{i}\frac{{\hat{b}}_{i}-{\hat{b}}_{j}}{{\hat{a}}_{i}-{\hat{a}}_{j}}+{\hat{b}}_{i}\;\mathrm{or}\;{\hat{y}}_{c}=-{\hat{a}}_{j}\frac{{\hat{b}}_{i}-{\hat{b}}_{j}}{{\hat{a}}_{i}-{\hat{a}}_{j}}+{\hat{b}}_{j} \end{array}\right..$$

Then, the building outline was established as completely as possible, which can been see in Figure 12b. Taking the established outline $\hat{f}(x)$—which was expressed by Equation (4) and whose parameters are shown in Table 4 as a cluster center—and using the means of yes (Y) or no (N) voting, the point cloud P was determined:(9)$$P=\left\{ \begin{array}{l} 1\;\;\;if\left|\hat{f}(x)-y\right|<\varepsilon \\ 0\;\;\;if\left|\hat{f}(x)-y\right|>\varepsilon \end{array}\right.$$

The determined point cloud is shown in Figure 13a. Then, through a coordinate transformation, we obtained the cloud point in the ground range coordinate, which is shown in Figure 13b.

From Figure 13b, we can see that the inverted point cloud (green dots) exactly lay on the building outline (red line). When we initialized different values, the learning results of the neural networks were also different. Compared with the method of Zhou et al. [5], Figure 14 and Figure 15 show the results obtained with our improved method for two initializations. Table 5 and Table 6 shows the relative precise nature of the TomoSAR point cloud after several iteration times of the neural network. From these two experiments, we can see that our method improved the precision of the estimated building outline. 

## 5. Conclusions

A machine-learning-based method was proposed for filtering these interferences and training the continuous outline of a building in cloud points. The flow chart of our method is shown in Figure 16 and steps of the proposed data processing method are as follows. First, a Hough transform is made use of to detect the lines on the tomography plane in TomoSAR point clouds. These detected lines lie on the outline of the building, but they are broken due to the density variation of point clouds. Second, the detected lines using a Hough transform are grouped as a date set for training the building outline. Unsupervised clustering is utilized to classify the lines in several clusters. The cluster number is automatically determined using the unsupervised clustering algorithm, which means the number of straight segments of the building edge is obtained. The lines in each cluster are considered to belong to a same straight segment of the building outline. Then, within each cluster, which represents a part or a segment of the building edge, a repaired straight line is constructed. Third, between each two clusters or each two segments of the building outline, the joint point is estimated by extending the two segments. Therefore, the building outline is obtained as completely as possible. Finally, taking the estimated building outline as the clustering center, supervised learning algorithm is used to classify the building cloud point and the noise (or false targets), and then the building cloud point is refined. Then, our refined and unrefined data are feed into the neural network for building the 3-D construction. The comparison results show the correctness and the effectiveness of our improved method.

## Figures and Tables

**Figure 1 sensors-19-05378-f001:**
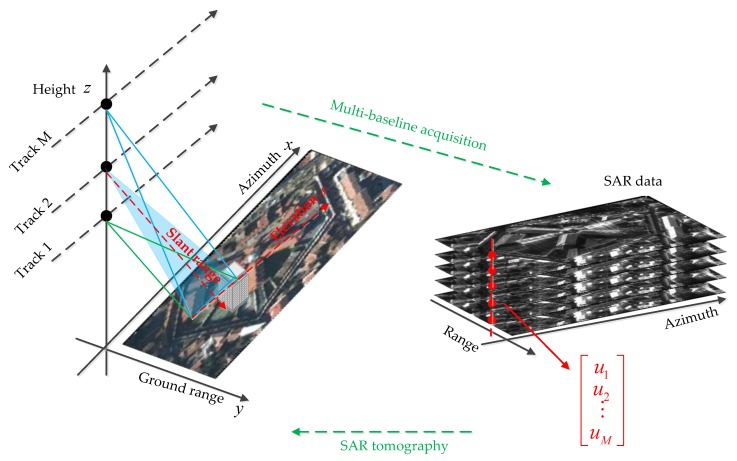
Synthetic aperture radar tomography (TomoSAR) model [21].

**Figure 2 sensors-19-05378-f002:**
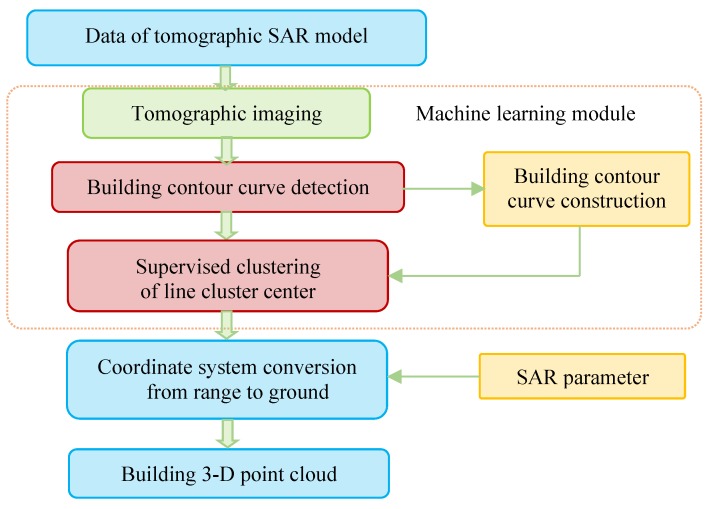
Tomographic synthetic aperture radar (TomoSAR) 3-D building reconstruction framework.

**Figure 3 sensors-19-05378-f003:**
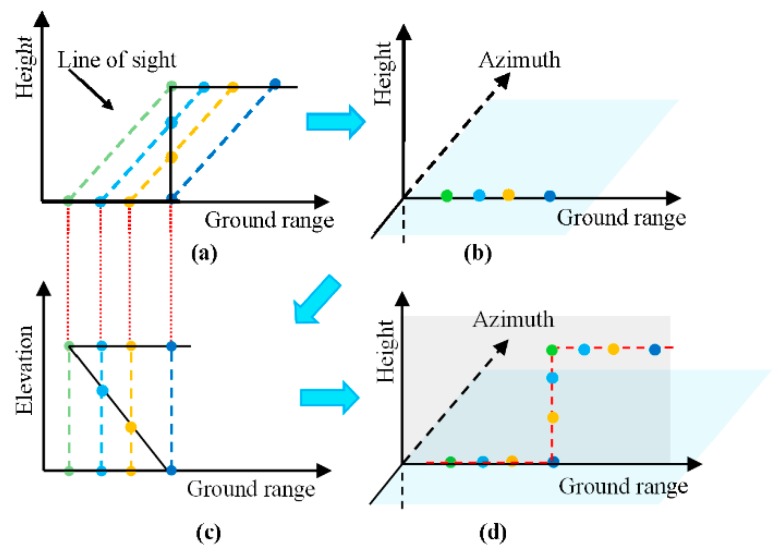
Building outline modification chain in the TomoSAR 3-D reconstruction. (**a**) Scattered layover principle. (**b**) Scattered layover results due to dropping into the same range-azimuth cell. (**c**) Scatters were inversed using tomographic imaging in elevation. (**d**) Scattered back to the original position using a coordinate conversion.

**Figure 4 sensors-19-05378-f004:**
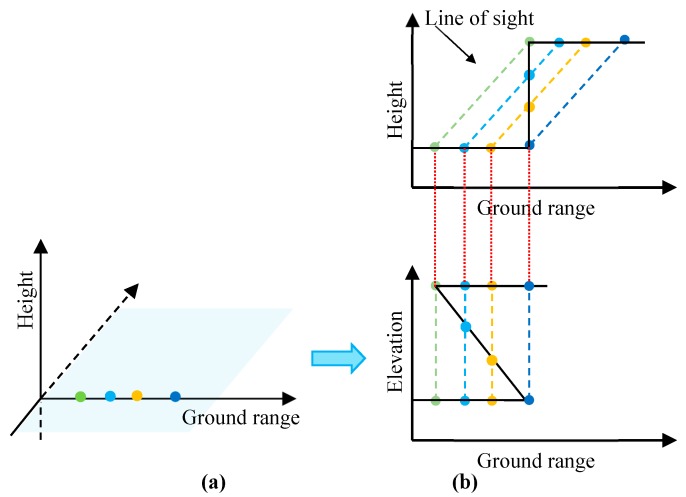
Compressive sensing imaging process of tomographic SAR, which completes a partial inversion process of the scattering points: (**a**) Scattered in a same range-azimuth cell. (**b**) Scatters inversed using tomographic imaging in elevation.

**Figure 5 sensors-19-05378-f005:**
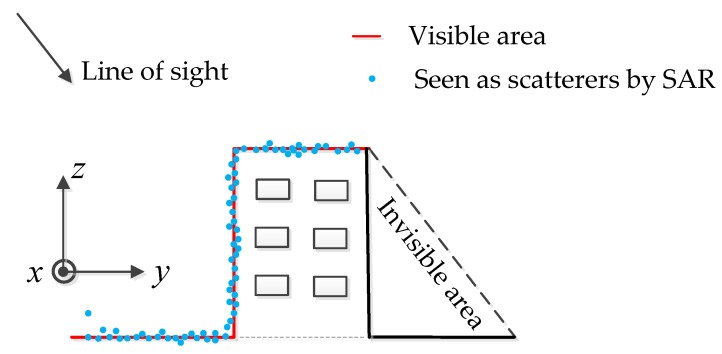
Schematic geometry of the 3-D building model. Red lines denote the “visible” area, while blue dots denote the scatterers [5].

**Figure 6 sensors-19-05378-f006:**
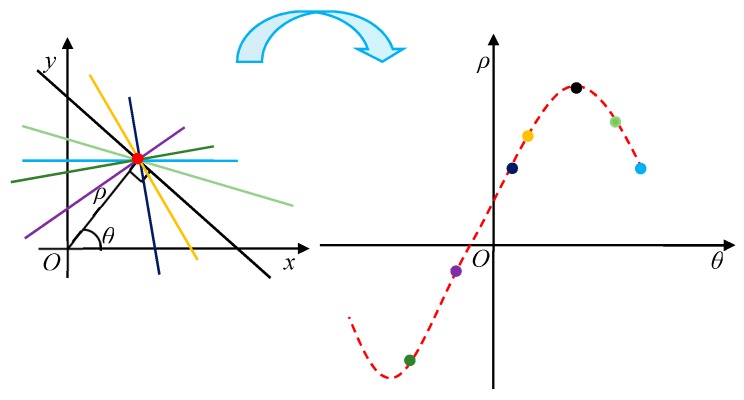
Relationship between (x,y) and (ρ,θ).

**Figure 7 sensors-19-05378-f007:**
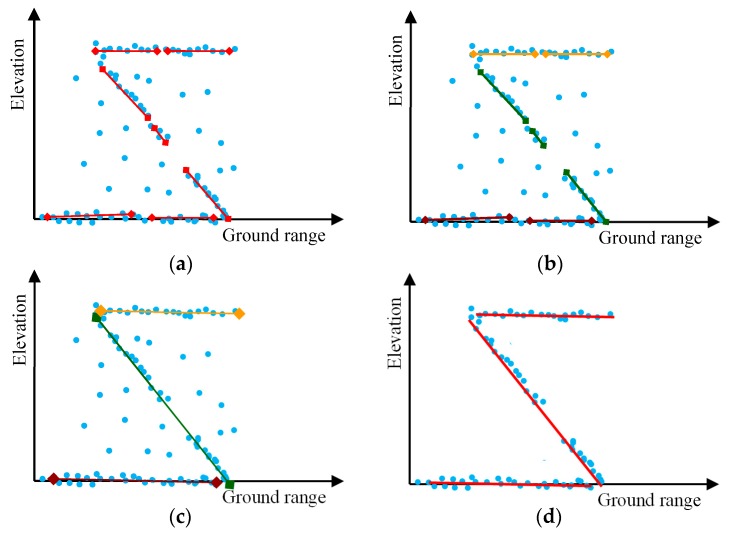
Building outline inversing chain. (**a**) Initial estimation of the building outline using the Hough transform. (**b**) K-means clustering of the parameters of the Hough transform. (**c**) Estimation of the continuous building outline with the result of unsupervised clustering. (**d**) Clustering the point cloud with regard to the line as the clustering center.

**Figure 8 sensors-19-05378-f008:**
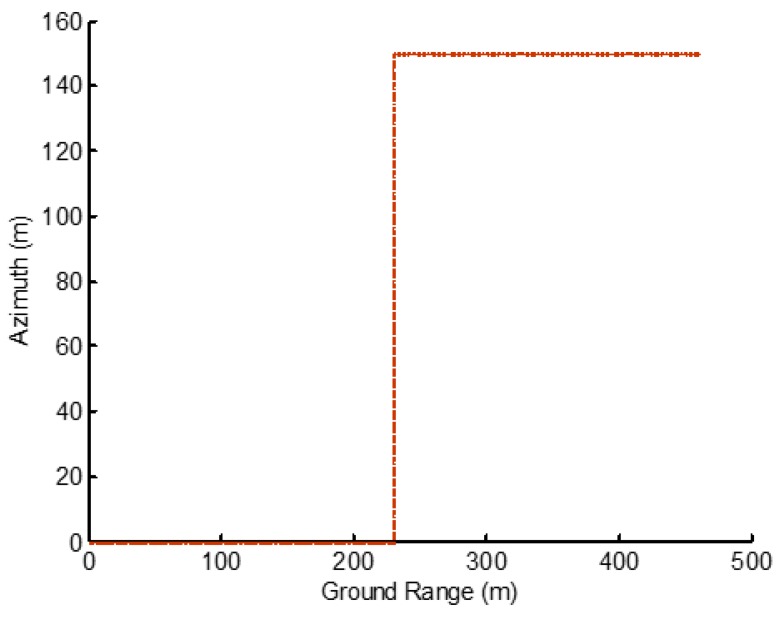
Simulated building scene whose height was 150 m.

**Figure 9 sensors-19-05378-f009:**
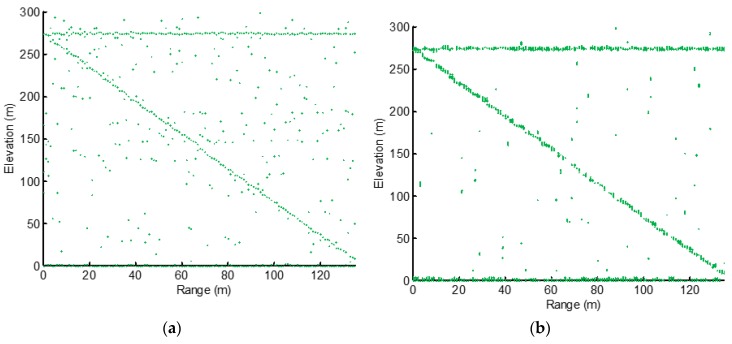
Tomographic imaging results. (**a**) Tomographic imaging result with orthogonal match pursuit (OMP). (**b**) Tomographic imaging result with regularized OMP (ROMP).

**Figure 10 sensors-19-05378-f010:**
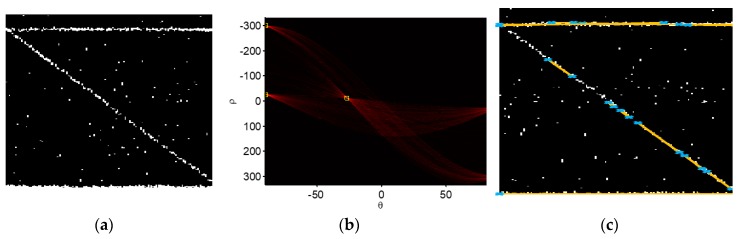
Hough transform line detection. (**a**) Binary image of point cloud. (**b**) Hough transform result. (**c**) Hough line detection result with the Hough transform.

**Figure 11 sensors-19-05378-f011:**
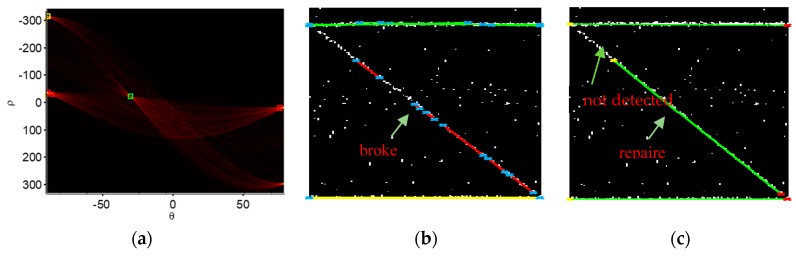
K-means clustering for line classification. (**a**) Features (ρ,θ) were grouped. (**b**) Lines were grouped into three clusters (red cluster, green cluster, and yellow cluster). (**c**) Repaired segments were constructed but they were still broken.

**Figure 12 sensors-19-05378-f012:**
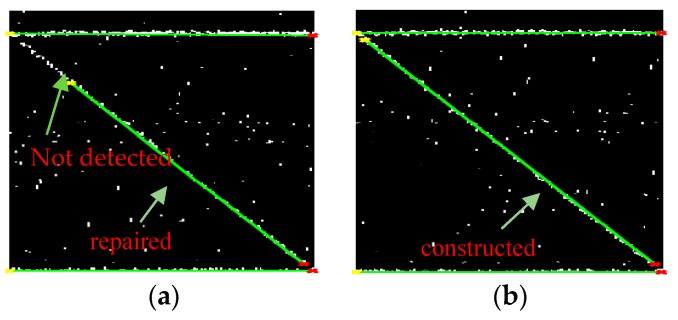
Building outline is constructed. (**a**) A repaired outline was constructed but it was broken because of the detection error. (**b**) A constructed outline.

**Figure 13 sensors-19-05378-f013:**
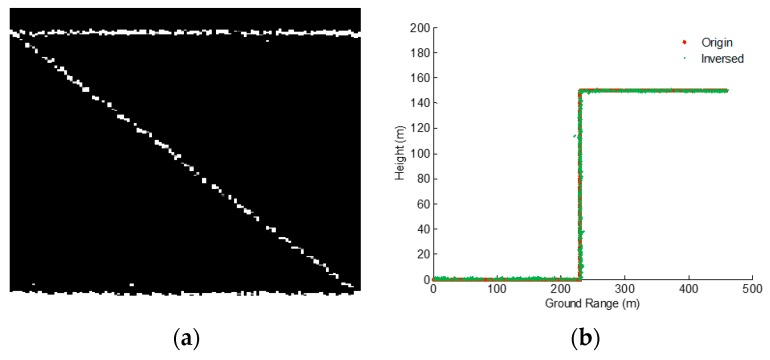
Supervised clustering result with the lines as a clustering center. (**a**) Clustering for filtering the noise or false targets. (**b**) Projecting the cloud points to ground range.

**Figure 14 sensors-19-05378-f014:**
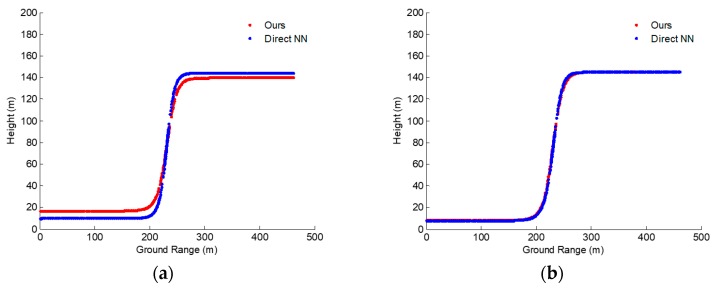
Construction results with our improved method (red dot line) and the method of reference (blue dot line) [5]. (**a**) Results after 300 iterations. (**b**) Results after 400 iterations. (**c**) Results after 500 iterations. (**d**) Results after 900 iterations. NN: neural network.

**Figure 15 sensors-19-05378-f015:**
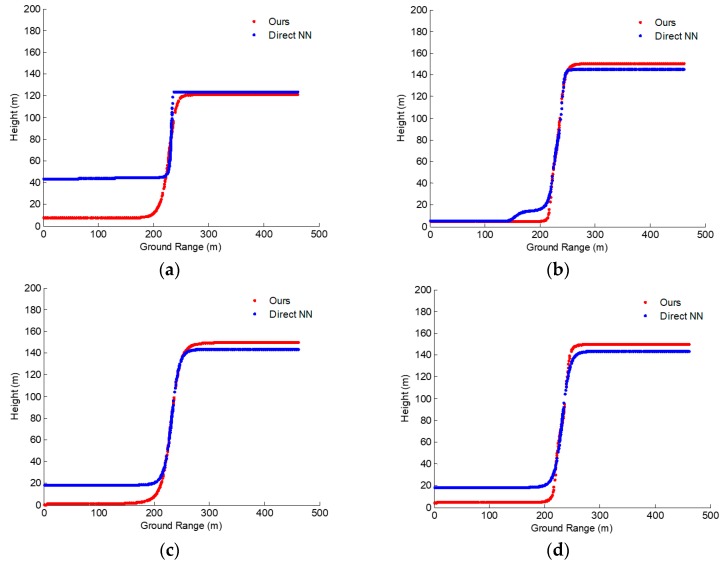
Construction results with our improved method (red dot line) and the method of reference (blue dot line) [5]. (**a**) Results after 300 iterations. (**b**) Results after 400 iterations. (**c**) Results after 420 iterations. (**d**) Results after 430 iterations.

**Figure 16 sensors-19-05378-f016:**
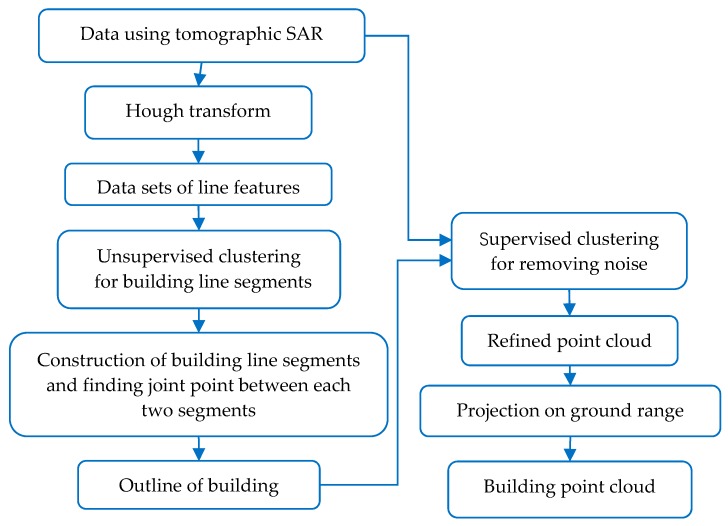
Flow chart of our data processing.

**Table 1 sensors-19-05378-t001:** The estimated parameters of lines.

No.	Start Pixel	End Pixel	θ (°)	ρ (Pixels)	Group Mark (Color)
1	(29,84)	(43,111)	−27	−13	red
2	(64,153)	(67,160)	−27	−13	red
3	(71,166)	(76,177)	−27	−13	red
4	(81,186)	(105,234)	−27	−13	red
5	(108,239)	(118,260)	−27	−13	red
6	(121,265)	(135,293)	−27	−13	red
7	(1,27)	(139,27)	−90	−26	green
8	(1,301)	(139,301)	−90	−300	yellow
9	(31,24)	(44,24)	−87	−21	green
10	(49,25)	(105,27)	−87	−21	green
11	(110,28)	(139,29)	−87	−21	green
12	(1,29)	(96,24)	87	28	green

**Table 2 sensors-19-05378-t002:** Simulation parameters of the SAR system.

Parameter	Symbol	Value
Wavelength	λ	0.0311 m
Incidence angle	θ	33.1284°
Center range	r	603638.971 m
Bandwidth	$B_{wr}$	150 MHz
Range resolution	$\rho_{r}$	1 m
Number of images	M	24

**Table 3 sensors-19-05378-t003:** Baseline parameters.

No.	Length (m)	Inclination (°)	No.	Length (m)	Inclination (°)
0	0	any	12	792.207	32.3227
1	35.712	33.9964	13	800.280	33.77537
2	97.540	33.4859	14	814.724	33.5181
3	126.987	33.6439	15	849.129	32.7626
4	141.886	33.6147	16	905.792	34.0288
5	157.613	32.9129	17	913.376	32.1973
6	278.498	33.4394	18	915.736	33.0059
7	421.761	32.4708	19	957.167	32.8915
8	485.376	33.5405	20	957.507	33.6594
9	546.882	32.1921	21	959.492	33.7188
10	632.359	32.6822	22	964.889	32.5021
11	655.741	32.2207	23	970.593	33.1079

**Table 4 sensors-19-05378-t004:** The estimated parameters of lines.

Index i	${\hat{a}}_{i}$	${\hat{b}}_{i}$
1	1.96	28.63
2	0.00	26.00
3	0.00	300.00

**Table 5 sensors-19-05378-t005:** Relative precision of TomoSAR height point cloud.

Iterations	Method of Reference [5]	Our Method
300	30.89	30.42
400	30.81	28.94
500	30.79	28.56
900	30.79	28.52

**Table 6 sensors-19-05378-t006:** Accuracy of TomoSAR height point cloud.

Iterations	Method of Reference [5]	Our Method
300	39.63	33.52
400	30.58	28.20
420	31.46	28.48
430	31.46	28.00

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
