# Peer review of "Hough Transform and Clustering for a 3-D Building Reconstruction with Tomographic SAR Point Clouds"

_sensors, 2019, doi:10.3390/s19245378_

Round 1

Reviewer 1 Report

Although paper tackles an important issue for the possible readers of the journal paper can not be published in its current form. My reasonings are as below:

Novelty is very limited The fallowed approach is a simple step-by-step solution In experiments, there is no real-world data set In experiments, the employed synthetic data set is not sufficient for showing the performance of the method There are almost no comparison studies  English needs serious attention

Author Response

This manuscript mainly focuses on the procedure and method of data processing because the quality of data processing directly affects the final result. The novelty of this work is exactly that we introduce Hough transform and clustering method to obtain the function of the building outline. Through this process, TomoSAR point cloud data is refined. No real data is used for the experiments because our laboratory currently lacks real data. In fact, when real data is used for verification, the selected data should also meet the basic conditions just as the simulation condition of this work. So here we considered that the simulation data plays the same role as real data. Comparison studies are added up to verify our approach. This manuscript mainly compares with the method of reference 5. In reference 5, TomoSAR data is directly use to establish the build outline. However, our method is to filtering TomoSAR data by Hough transform and clustering method before establishing the build outline. For English language and style, my coauthors and I invited a native English speaking colleague to help us modify the grammar which mainly is focused on the singular and plural forms of nouns and verbs. And he also modifies the manuscript according to the expression habits and language sequence.

Reviewer 2 Report

T

This paper showed the system for filtering interferences and training the continues outline of building in cloud points. The results o this paper are clear. However, this paper did not have a comparison with previous reserach to show the effectiveness of this proposed system.

1)Please show a comparison with previous research.

2)Please show the future development of this research and its effective application fields, in more detail.

Author Response

Comparison studies are added up to verify our approach. This manuscript mainly compares with the method of reference 5. In reference 5, TomoSAR data is directly use to establish the build outline. However, our method is to filtering TomoSAR data by Hough transform and clustering method before establishing the build outline. About the future trend of this research and its effective application fields, we added some contents in Section 1.

Round 2

Reviewer 1 Report

Although the proposed method is interesting for the potential reader of the journal study itself has limited novelty and paper is not well written. My concerns are as below

- In abstract, improve the sentence '... and training the continuous outline of building in cloud points'.
- In the abstract, 'The data processing method are as follows.' --> 'Steps of the proposed data processing method are as follows.'
- Please put spaces between words and references, for example 'areas[5]' should be 'areas [5]'.
- Please put spaces between sentences, for example 'in the future.In urban remote' should be 'in the future. In urban remote'.
- 'areas. the point cloud' --> 'areas. The point cloud'
- 'sensing process.. Recurrent' --> 'sensing process. Recurrent'
- 'produce3-D point clouds' --> 'produce 3-D point clouds'
- 'compressive sensing(CS)' --> 'compressive sensing (CS)'
- 'Figure 7(red line segments).' --> 'Figure 7 (red line segments).'
- 'Figure 7(b)(each color' --> 'Figure 7(b) (each color'
- 'detected lines(red line' --> 'detected lines (red line'
- '3.2. unsupervised and supervised clustering' --> '3.2. Unsupervised and supervised clustering'
- Section 3.2 (unsupervised and supervised clustering) is not clear. What is the employed unsupervised clustering method? what are its parameters? How ground-truth is prepared for supervised learning? what are the details for the learning procedure? there are a lot of unclear things in this section.
- Authors do not have real data so they preferred using synthetic data. Although this is not completely sufficient it can be assumed to be reasonable if the synthetic data is very representative, ie. buildings with different characteristics, isolated and nearby buildings, buildings under and near the trees, etc. However, in this study author used simple synthetic data, which is to me is not convincing and does not present the power and weakness of the proposed method.
- 'Then ,the building' --> 'Then, the building'
- The use of neural networks as the last step of the proposed method is not clearly explained thus does not sound.

Author Response

In abstract, improve the sentence '... and training the continuous outline of building in cloud points'. We modify it as: 'In this paper, Hough transform is adopted to detect the outline of a building; however, the obtained outline of a building with Hough transform is broken on one hand and on the other hand some of these broken lines belong to the same segment of a building outline, but the parameters of these lines are slightly different. These problems will lead to that a segment of a building outline is represented by multiple different parameters in Hough transform. Therefore, unsupervised clustering method is employed for clustering these line parameters. The lines gathered in the same cluster are considered to correspond to a same segment of a building outline. In this way, different line parameters corresponding to a segment of a building outline are integrated into one and then the continuous outline of building in cloud points is obtained. ' In the abstract, 'The data processing method are as follows.' According to the suggestion we modify it as 'Steps of the proposed data processing method are as follows.' Please put spaces between words and references, for example 'areas[5]' should be 'areas [5]'. We put spaces between words and references for all similar cases. Please put spaces between sentences, for example 'in the future.In urban remote' should be 'in the future. In urban remote'. We put spaces between sentences for all similar cases. 'areas. the point cloud', we modify it as 'areas. The point cloud'. And we modify all similar cases. 'sensing process.. Recurrent', we modify it as 'sensing process. Recurrent'. And we modify all similar cases. 'produce3-D point clouds', we modify it as 'produce 3-D point clouds'. And we modify all similar cases. 'compressive sensing(CS)', we modify it as 'compressive sensing (CS)'. And we modify all similar cases. 'Figure 7(red line segments).', we modify it as 'Figure 7 (red line segments).'. And we modify all similar cases. 'Figure 7(b)(each color', we modify it as 'Figure 7(b) (each color’. And we modify all similar cases. 'detected lines(red line', we modify it as 'detected lines (red line'. And we modify all similar cases. '3.2. unsupervised and supervised clustering', we modify it as '3.2. Unsupervised and supervised clustering'. And we modify all similar cases. Section 3.2 (unsupervised and supervised clustering) is not clear. What is the employed unsupervised clustering method? what are its parameters? How ground-truth is prepared for supervised learning? what are the details for the learning procedure? there are a lot of unclear things in this section. Here is our response: 1) The employed unsupervised clustering method. The unsupervised clustering method that we employed is K-mean algorithm. Because in a tomographic plane, we don't know how many segments there are. When we don't know how many clusters to gather and we hope to get effective clustering results, the solution to this problem is called unsupervised method. 2) about the parameters. After Hough transform, some line parameters are obtained. These line parameters form the data set and we will use K-means to cluster this data set. 3) How ground-truth is prepared for supervised learning? After the outline is obtained, the points close to the outline should be left, and the points far from the outline should be removed as noise or false targets. Based on this outline, point clouds are divided into two categories: one is the outline scatterers of buildings, the others are noise, so these is called supervised learning because we cluster point clouds with the outline as reference. 4) what are the details for the learning procedure? About the details for the learning procedure, please see in manuscript. Authors do not have real data so they preferred using synthetic data. Although this is not completely sufficient it can be assumed to be reasonable if the synthetic data is very representative, ie. buildings with different characteristics, isolated and nearby buildings, buildings under and near the trees, etc. However, in this study author used simple synthetic data, which is to me is not convincing and does not present the power and weakness of the proposed method. Single building can prove the ability of noise removal of this method, and then improve the estimation results of building outline in TomoSAR point cloud. Buildings with different characteristics, here we only discuss the outline shape characteristics. In the tomographic plane, most of the outline of buildings are made up of several straight line segments. About nearby buildings, buildings under and near the trees and so on, these things are invisible in TomoSAR point cloud, we only focus on the targets visible in the point cloud. So an isolated building can be used to verify methods. 'Then ,the building', we modify it as'Then, the building'. And we modify all similar cases. The use of neural networks as the last step of the proposed method is not clearly explained thus does not sound. We use the method of neural network, which is reference [5]. Through the previous improvement method, we refine our data, which makes the effect of using neural network better.

Reviewer 2 Report

This paper showed the system for filtering interferences and training the continues outline of building in cloud points.  The results o this paper are clear. However, please correct the following.

The Figure 5 of this paper is exactly the same as Figure 3 in Ref. [5]. Even if it is the same journal, please correct Figure 5 to another figure.

Author Response

Figure 5 of this paper is exactly the same as Figure 3 in Ref. [5]. Even if it is the same journal, please correct Figure 5 to another figure.

Our response: Thank you very much for your comments. This question has been revised in my manuscript. Your review is my best learning opportunity.

Round 3

Reviewer 1 Report

Authors improved the paper as suggested other than real-world experiments which is not done since finding such data is very challenging (as claimed by the authors). Although scientific novelty of the paper is limited it addresses an important problem in a simple but scientifically sounding way. Therefore, paper can be published in its current form.